# Effect of EDTA Gel on Residual Subgingival Calculus and Biofilm: An In Vitro Pilot Study

**DOI:** 10.3390/dj11010022

**Published:** 2023-01-09

**Authors:** Charles M. Cobb, Stephen K. Harrel, Donggao Zhao, Paulette Spencer

**Affiliations:** 1Department of Periodontics, School of Dentistry, University of Missouri-Kansas City, Kansas City, MO 64108, USA; 2Department of Periodontics, Texas A & M, College of Dentistry, Dallas, TX 75246, USA; 3Electron Microscope Laboratory, School of Dentistry, University of Missouri-Kansas City, Kansas City, MO 64108, USA; 4Mechanical Engineering and Director, Institute for Bioengineering Research, University of Kansas, Mechanical Engineering, Lawrence, KS 66045, USA

**Keywords:** dental calculus, biofilms, edetic acid, scanning electron microscopy, energy dispersive spectroscopy

## Abstract

Background: Residual calculus, following scaling and root planing (SRP), is associated with persistent inflammation and the progression of periodontitis. This study examined the effects of a 24% neutral ethylenediaminetetraacetic acid (EDTA) gel on subgingival calculus and biofilms. Methods: Eleven single-rooted teeth extracted because of severe periodontal disease were randomly assigned to the following treatment groups: (1) three teeth served as untreated controls; (2) three teeth were treated by scaling and root planing (SRP) only; and (3) three teeth were treated by SRP + EDTA. The remaining two teeth, one SRP only and the other SRP + EDTA were designated for energy-dispersive X-ray spectroscopy (EDS) analysis. EDTA gel was placed on the SRP surface for 2 min and then burnished with a sterile cotton pellet. Results: SRP + EDTA treated specimens exhibited severely damaged biofilm and the disruption of the extracellular polymeric matrix. EDS scans of the smear layer and calculus featured reductions in the Weight % and Atomic % for N, F, Na, and S and increases in Mg, P, and Ca. Conclusions: A 25% neutral EDTA gel was applied after SRP severely disrupted the residual biofilm and altered the character of dental calculus and the smear layer as shown by reductions in the Weight % and Atomic % for N, F, Na, and S and increases in Mg, P, and Ca.

## 1. Introduction

The introduction of minimally invasive surgical techniques combined with high-resolution dental videoscopes, when used to treat periodontitis, resulted in the discovery of root surface features not previously reported, i.e., microgrooves [1,2] and microislands of the calculus [3]. The microislands are embedded in cementum and represent residual deposits of calculus following scaling and root planing (SRP). Microislands of calculus are not visible with 3.5× surgical loupes but can be identified by fluorescence when using a 655 nm diode laser [3] and visualized at 40× using a videoscope [3,4]. The contributions of microislands of calculus to persistent inflammation in the periodontal pocket soft tissue wall or their function as a possible nidus for the deposition of new layers of calculus and biofilms are unknown. In vitro studies have shown that micro-sized, needle-shaped hydroxyapatite particles derived from calculus are toxic to epithelial cells and macrophages [5,6,7], causing the affected cells to develop inflammasomes and secrete IL-1ß: an important mediator of inflammation. Thus, it appears that any residual calculus, regardless of the amount and size of the deposit, may exert a detrimental effect on the resolution of inflammation and healing of the periodontal wound [8].

Rohanizadeh and Legeros [9] reported that hydroxyapatite crystals of calculus and cementum can bind to one another. Further, fractographic analysis has demonstrated the calculus-cementum attachment is stronger than the cohesive strength within calculus [10]. These findings may explain the incomplete removal of calculus deposits following SRP. It may also explain the findings of Harrel et al. [3] that microislands of calculus remain following “definitive” SRP.

A previous study reported that the use of a 24% EDTA gel following SRP appeared to significantly alter the physical properties of the microislands of calculus [3]. Thus, the purpose of this in vitro study was to investigate the effects of 24% neutral (pH 7.4) EDTA gel, used as an adjunct to SRP, on residual calculus and biofilms remaining on periodontally diseased teeth following SRP. To this end, scanning electron microscopy (SEM) and energy-dispersive X-ray spectroscopy (EDS) were used to determine changes in the physical structure and elemental composition.

## 2. Materials and Methods

### 2.1. Specimens

Specimens consisted of eleven extracted single-rooted teeth that had been extracted because of severe periodontal disease. All specimens met the following inclusion criteria: (1) no caries involving the root surface; (2) visible calculus on the diseased root surface; (3) no visible root fractures or anatomical abnormalities involving the root surface; (4) a root surface that was relatively flat and presented an adequate area for treatment. Immediately upon extraction, the specimens were rinsed free of blood using ice-cold 0.9% saline, assigned a code number, and stored in individual bottles containing 10% neutral buffered formalin until treatment. Immediately prior to treatment, specimens were rinsed with sterile water.

No patient identifiers were associated with the specimens. The specimen teeth would otherwise be discarded, and thus, this study is not considered human-subject research. This interpretation was verified by the Texas A & M University, Division of Research (IRB2022-0458), which determined that “the proposed activity is not research involving human subjects as defined by DHHS and FDA regulations”.

### 2.2. Treatment

Teeth were randomly selected and assigned to the following treatment groups: (1) three teeth served as untreated controls; (2) three teeth were treated by SRP only; and (3) three teeth were treated by SRP + EDTA. The remaining two teeth, one SRP only and the other SRP + EDTA were designated for EDS analysis.

The treated root surface area was approximately 4 × 4 mm. The root surface with intact calculus was scaled and root planed with an ultrasonic scaler (Odontoson Dental Ultrasonic Scaling Unit, Goof, Denmark) and new Gracey curettes (Hu-Friedy Mfg. Co., LLC, Chicago, IL, USA).

SRP was performed using standard 3.5× surgical loupes and an attached surgical headlight. Using routine clinical procedures, the treatment area was scaled with an ultrasonic scaler until most visible calculus was removed and then hand scaled with new Gracey curettes until no visible calculus remained when observed with the 3.5× surgical loupes. The endpoint for SRP was no clinically observable calculus. All SRP was performed by a periodontist with 45+ years of clinical and teaching experience.

For specimens treated with EDTA, a commercially available 24% solution of neutral (pH 7.4) EDTA gel (PrefGel™, Straumann, Basel, Switzerland) was placed on the scaled root surface for 2 min and then burnished with a sterile cotton pellet. The EDTA was then removed from the root surface by rinsing it with distilled water. Following treatment and rinsing, the specimens were stored in a 10% neutral buffered formalin solution until processed for examination by SEM and EDS.

### 2.3. Scanning Electron Microscopy (SEM)

Specimen preparation for SEM (Philips XL30 FEG-ESEM scanning electron microscope, FEI Corp., North America NanoPort, Hillsboro, OR, USA) examination consisted of: (1) a thorough rinse in sterile water; (2) dehydration in a series of graded ethyl alcohol solutions (33–100%); (3) specimens mounted on aluminum stubs and stored in a desiccator until examined by SEM; and (4) immediately prior to SEM examination, specimens were sputter coated with approximately 200 Å of gold-palladium.

Specimens were examined at a variety of magnifications by a single investigator (CMC) who was blinded as to the treatment. The SEM was operated at an accelerating voltage of 15 kV with a working distance of 10 or 20 mm and a 40 μm spot size.

The 4 × 4 mm treated root surface area was first scanned at 40× magnification to identify residual calculus deposits and other features of interest. Following the initial examination, multiple areas of interest were then assessed at a variety of higher magnifications, ranging from 200× up to 15,000×. For the three SEM specimens, an aggregate total root surface and total root surface area of 48 mm^2^ were assessed, with 12 sites per treated root surface being examined, i.e., 36 sites.

### 2.4. Energy-Dispersive X-ray Spectroscopy (EDS)

The SEM was fitted with EDS [EDAX APEX Advanced, METEK Materials Analysis Division. AMETEK, Inc., Mahwah, NJ, USA] capability allowed for the identification and characterization of the elemental composition of selected sites on the samples. The two specimens selected for EDS were rinsed in deionized water for 30 min, refreshing the water every 5 min, and then stored in deionized water until use. Specimens were then sputter coated for 30 s with carbon. Parameters for the EDS scan were an accelerating voltage of 15 kV, a working distance of 10 mm, a 150 nm beam size, a takeoff angle of 35°, and a magnification of 39×. Semi-quantitative data (weight % and atomic %) were generated using a line scan mode. Counts along the scan line were 3576 (SRP only) and 4105 (SRP + EDTA). Elements of interest were nitrogen (N), oxygen (O), fluorine (F), sodium (Na), magnesium (Mg), phosphorus (P), sulfur (S), and calcium (Ca).

## 3. Results 

### 3.1. SEM Examination

Within the 48 mm^2^ surface area of SRP only and SRP + EDTA-treated root surfaces, it was estimated that <10% of the area exhibited residual calculus and/or intact biofilm deposits. The three untreated specimens were consistent in character, featuring a rough topography due to scattered deposits of the calculus and biofilm. Contributing to the irregular root topography were numerous areas of cavitation-like defects, also known as resorption bays or lacunae (Figure 1). The cavitational defects were frequently filled with a developing biofilm consisting of short and medium-length rods and coccoid microbes. The examination of calculus deposits revealed a lush surface layer of biofilm with an abundant extracellular polymeric matrix (EPM) (Figure 2 and Figure 3).

SRP-only specimens were characterized by a parallel pattern of instrumentation streaks and a smear layer (Figure 4). At high magnification, the smear layer appeared to consist mostly of bacteria and cementum debris. In areas lacking a smear layer, it was common to find clusters of bacteria for varying morphotypes, e.g., cocci, rods of various lengths, and an occasional spirochete.

Specimens treated by SRP + EDTA also exhibited instrumentation marks and a smear layer (Figure 5). However, in EDTA-treated specimens, the smear layer appeared “smudged” with little debris, undoubtedly due to the EDTA application technique, i.e., gentle burnishing with a cotton pellet. Examination at high magnification revealed sites of residual calculus and biofilm in which bacteria were dramatically altered, with both rods and cocci having ruptured the cell walls, exposing an empty interior (Figure 6 and Figure 7) and a noticeably depleted EPM (Figure 6, Figure 7 and Figure 8).

### 3.2. EDS Analysis

Comparison of EDS scans (Figure 9, Figure 10, Figure 11 and Figure 12) revealed a decreased concentration (Weight % and Atomic %) of the following elements: N, F, Na, and S in the SRP + EDTA specimen vs. the SRP-only specimen. There was an increase in the following elements: Mg, P, and Ca in the SRP + EDTA vs. SRP only. Oxygen levels were nearly equal in both specimens, i.e., 49.1 Weight % for SRP only vs. 49.6 Weight % for SRP + EDTA. These results represent semi-quantitative data with analysis uncertainty values (error of measurements) of 10.70% (Figure 13, SRP only) and 11.86% (Figure 14, SRP + EDTA). 

In comparing the weight % values for the elements of interest in the SRP-only (Figure 13) vs. SRP + EDTA specimens (Figure 14), the SRP + EDTA specimen showed a 69% decrease in N (18.5 vs. 5.8), a 35% decrease in F (2.3 vs. 1.5), a 45% decrease in Na (2.3 vs. 1.3), and a 69% decrease in S (1.6 vs. 0.5). In contrast, there was a 36% increase in Mg (0.7 vs. 1.1), a 33% increase in P (8.5 vs. 12.9), and a 61% increase in Ca (16.6 vs. 27.0) in the SRP + EDTA specimen.

## 4. Discussion

SEM examination revealed a consistent finding of the root’s surface cavitation-like defects (also known as lacunar defects). Such defects present a safe harbor for the development of biofilm, and without complete ablation, these sites will contribute to the infection of the periodontal pocket.

The SEM examination of residual calculus deposits in both SRP-only and SRP + EDTA specimens showed an abundant surface coat of biofilm. The EPM-surrounding individual bacteria were easily visualized in untreated and SRP-only specimens (Figure 3). The application of EDTA-damaged exposed bacteria disrupted the structure of the EPM (Figure 6, Figure 7 and Figure 8). This type of microbial damage has been previously reported following the subgingival use of tetracycline fibers [11]. EDTA treatment altered the EPM—this is an important observation as the EPM protects microbes from desiccation, biocides, and antibiotics [12].

There is substantial evidence that EDTA is an antimicrobial and antibiofilm agent. EDTA chelates Ca, Mg, Zn, and Fe and disrupt the cell walls of bacteria, and destabilizes biofilms [13,14]. EDTA can also decrease the divalent cation concentration of the EPM, thereby increasing water solubility and facilitating antimicrobial penetration [13]. Further, low concentrations of EDTA have been shown to prevent biofilms by inhibiting the adhesion of bacteria and reducing microbial colonization and proliferation [15,16]. Partially due to the antimicrobial nature of EDTA and the etching effect on mineralized surfaces, a 24% EDTA gel has been used as a surface modifier in periodontal root coverage surgeries, e.g., coronally advanced flaps, subepithelial connective tissue grafts, modified coronally advanced tunnel procedure, etc., with only modest benefits. It should be noted, however, regardless of any benefit derived from the use of EDTA, that clinical trials have reported no adverse effects. [17,18].

Both treated specimen groups exhibited a smear layer in areas of instrumentation (Figure 4 and Figure 5). It has been reported that SRP produces a 2.15 μm thick smear layer of microcrystalline debris, which is intimately associated with the root surface and is difficult to remove except when using demineralizing agents [19,20]. There was a difference in presentation, however, as the smear layer of EDTA-treated specimens was relatively homogenous. In contrast, the smear layer of SRP-only specimens was coarse and uneven and frequently contained clusters of bacteria. This latter observation reinforces the need for precise SRP as the incomplete removal of residual biofilms leads to the reinfection and progression of periodontal disease [4]. In this study, two mineralized tissues were the primary focus of the EDS analysis, i.e., root cementum and attached calculus. 

EDS analysis allows the in situ detection of the relative quantities of elements in specimens. With this technique, it is relatively quick to obtain compositional information with high spatial resolution. EDS does not provide the absolute quantities of elements in complex, heterogenous specimens but provides the relative proportion and distribution of elements in the area analyzed. Thus, in the SRP-only and SRP + EDTA, we see a Ca:P ratio of 1.95 and 2.0 by Weight %, respectively. This finding is in alignment with other reports in which EDS was used for the analysis of cementum [21,22].

Ca:P ratios should be interpreted with caution as the cementum of roots exposed to periodontal disease will exhibit variation in the Ca and P content from surface to surface and from tooth to tooth, even in the same subject. In spite of the potential for variation, Cohen et al. [21] reported that the Ca and P content was greater in roots exposed to pockets when compared to roots exposed to gingival recession. This observation is supported by the fact that roots exposed to pockets associated with periodontitis tend to be hypermineralized [23,24,25,26].

The EDS scan peak for O is related to organic materials, e.g., bacteria, collagen matrix, biofilm EPM, and so forth. Thus, one would not expect a large difference in the relative concentration between the two specimens. However, the N peaks differed by a factor of 3×; the relative concentration in SRP only being 18.5 vs. 5.8 in the SRP + EDTA specimen. The decrease in the N concentration reflects decreased proteins on the surface of the SRP + EDTA specimen and likely reflects a denatured bacterial EPM and smear layer [27].

The Na, F, and Mg EDS peaks may represent examples of ion substitutions in the calcium hydroxyapatite crystalline structure. For example, in the case of Mg, the substitution of Mg in the apatite structure, Ca_10_(PO_4_)_6_(OH)_2_, would yield Ca_10_Mg × (PO_4_)_6_(OH)_2_. Magnesium is also found in whitlockite, which is mainly associated with the pathologic mineralization of various tissues, including dental calculus. [28] This latter fact explains the localization of the Mg peaks to the calculus seen in Figure 11.

Other factors to consider are the localization of Ca and Mg in biofilms. Song and Leff [29] reported that biofilm surface colonization and depth increased with increasing Mg concentrations. The authors further reported that Mg increases the initial attachment of bacteria which, in turn, may alter subsequent biofilm formation and structure.

The Na peak, localized to calculus, may result from the same scenario as Mg. The present study confirms the findings of Bosshardt and Selvig [23] and Wirthlin et al. [30], who reported that root surfaces exposed to the oral environment due to a loss of attachment exhibit significant increases in Ca, Mg, P, and F. The present study found a concentration of 0.7% Mg in the SRP-only specimen, which increased to 1.1% in the SRP + EDTA-treated specimen (Figure 13 and Figure 14). Mg has been reported to occur at lower concentrations at the surface than in deeper cemental layers [22]. Thus, the increased Mg level in the EDTA-treated specimen may result from the exposure of deeper cementum layers due to an etching effect.

The S peak probably represents the sulfated glycosoaminoglycans and proteoglycans in the matrix of both cementum and biofilms. In this respect, it is interesting to note that the SRP + EDTA-treated specimen showed a 69% decrease in S vs. that of the SRP-only specimen, suggesting that EDTA disrupted the sulfated components of the extracellular matrix, particularly in biofilm. The disruption of the EPM is also noted in the SEM images of the SRP + EDTA specimens (Figure 6, Figure 7 and Figure 8).

Although the current investigation is a pilot study presenting descriptive results, limitations must be considered for the design of future efforts. The major limitation is the small sample size which does not allow for the statistical analysis of the EDS data. An additional consideration would be the expansion of the experimental design to include other biological surface modifiers for comparison to EDTA.

## 5. Conclusions 

A 25% neutral EDTA gel was applied after SRP severely disrupted the EPM and microbes found in subgingival biofilms. EDTA also alters the character of the smear layer and residual calculus manifested by reductions in the Weight % and Atomic % for N, F, Na, and S and increases in Mg, P, and Ca. The EDTA-mediated disruption of biofilm microbes and their EPM and the smear layer resulting from root surface instrumentation may enhance SRP and decrease periodontal inflammation.

## Figures and Tables

**Figure 1 dentistry-11-00022-f001:**
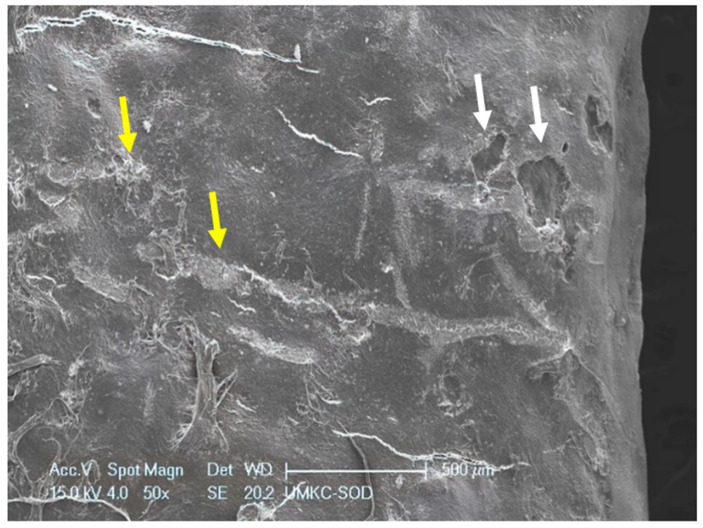
SRP-only treated specimen exhibiting deposits of calculus (**yellow arrows**) and several cavitation-like defects in root surface (**white arrows**). Original magnification of 40×; Bar = 500 μm.

**Figure 2 dentistry-11-00022-f002:**
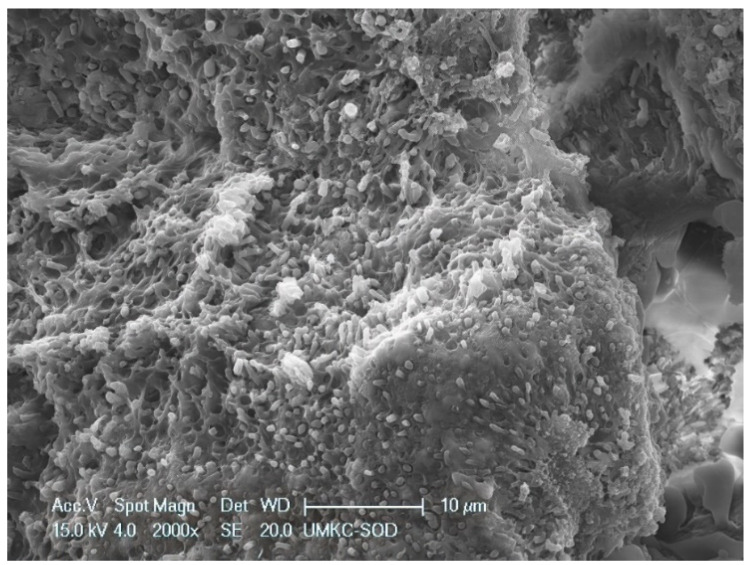
Untreated root surface showing calculus of a calculus shelf covered with biofilm. Note bacteria embedded in extra-cellular matrix of the lateral wall of the calculus shelf. Original magnification of 2000×; Bar = 10 μm.

**Figure 3 dentistry-11-00022-f003:**
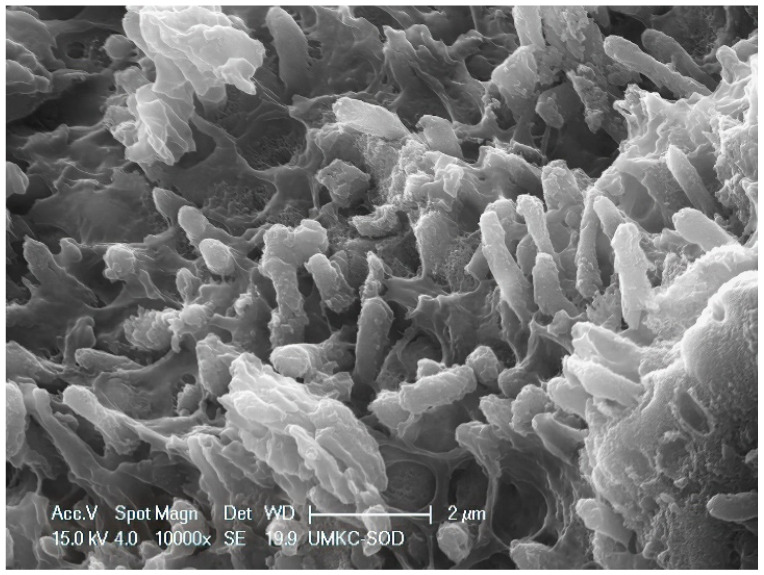
Untreated root surface at high magnification showing biofilm magnification view of biofilm microbes embedded in the extracellular matrix. In this view, most of the microbes are a short rod morphotype of ≈2 μm length. Original magnification of 10,000×; Bar = 2 μm.

**Figure 4 dentistry-11-00022-f004:**
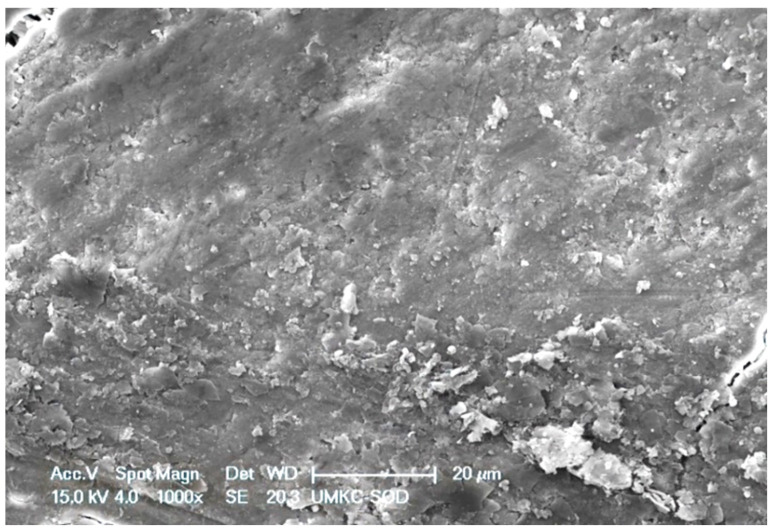
SRP-only treated specimen characterized by a smear layer consisting of loosely attached debris and a parallel pattern of instrumentation streaks. Original magnification of 1000×; Bar = 20 μm.

**Figure 5 dentistry-11-00022-f005:**
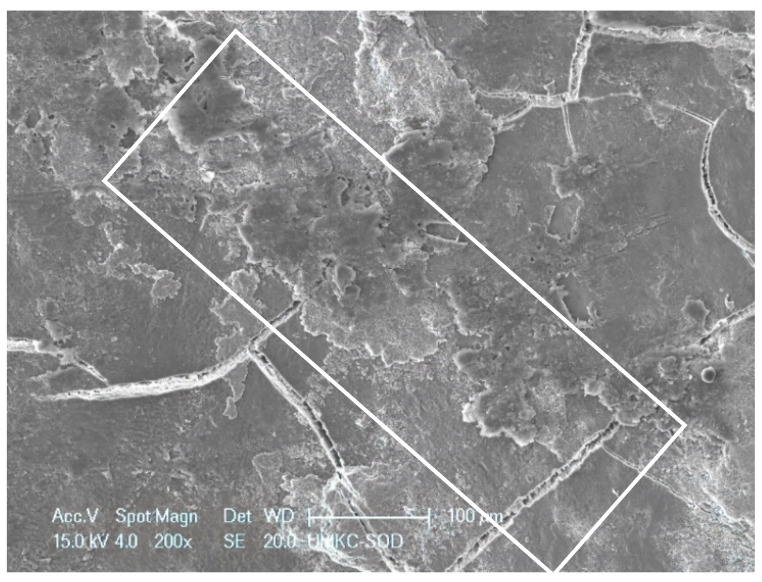
SRP + EDTA-treated specimen. Treated surface features a smearing or smudged appearance (**white rectangle**). Original magnification. 200×; Bar = 100 μm.

**Figure 6 dentistry-11-00022-f006:**
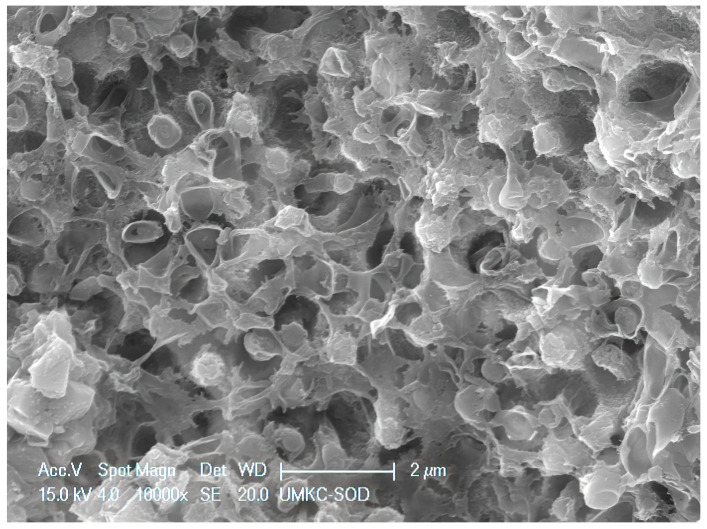
SRP + EDTA-treated specimen. High magnification view of a dense aggregate of coccoid bacteria showing the effect of the EDTA. Many of the microbes appear to have ruptured. Additionally, note the loss of extracellular matrix. Original magnification of 5000×; Bar = 5 μm.

**Figure 7 dentistry-11-00022-f007:**
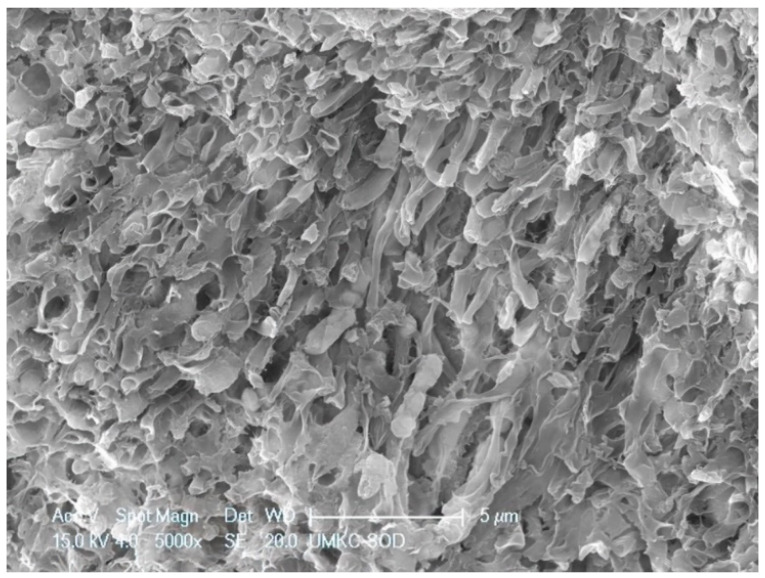
SRP + EDTA-treated specimen. High magnification view of a cluster of rod-shaped microbes showing loss of EPM, rupture of terminal ends of many of the microbes, and general loss of distinctive morphology. Original magnification of 5000×; Bar = 5 μm.

**Figure 8 dentistry-11-00022-f008:**
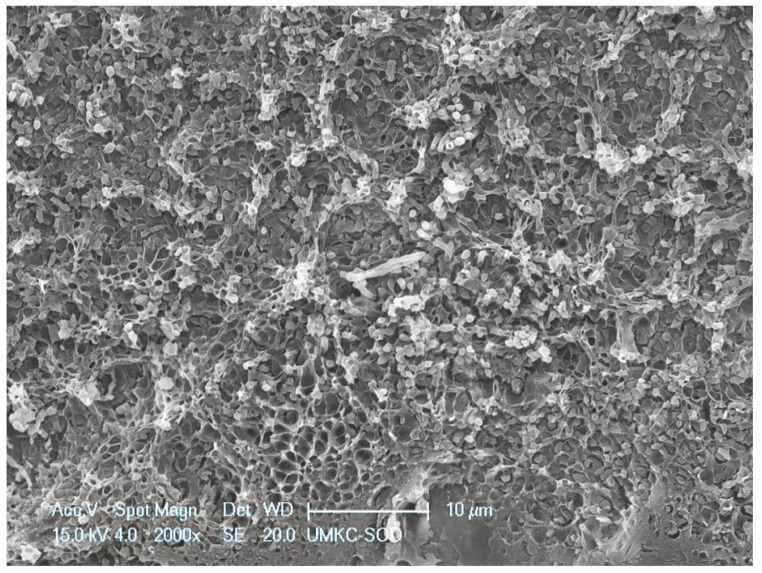
Area of residual biofilm following treatment with SRP + EDTA. Note the honeycomb appearance of the EPM indicating loss of structural integrity and loss of embedded microbes. Original magnification of 2000×; Bar = 10 μm.

**Figure 9 dentistry-11-00022-f009:**
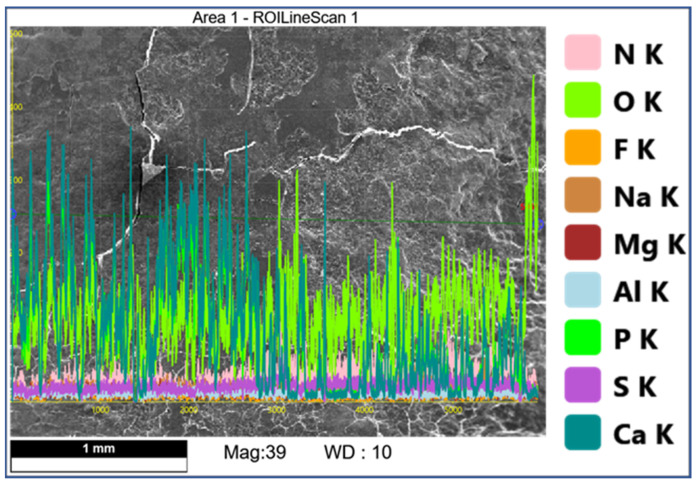
SEM of SRP-only specimen with overlay of color-coded EDS line scan showing peaks (from baseline) for F, Al, S, N, Ca, P, and O. Mg and N are present but in low concentrations and are barely discernable. The scan traversed a line of ≈3 mm in length which included the smear layer resulting from SRP. Original mag. 39×; Bar = 1 mm. Working distance of 10 mm.

**Figure 10 dentistry-11-00022-f010:**
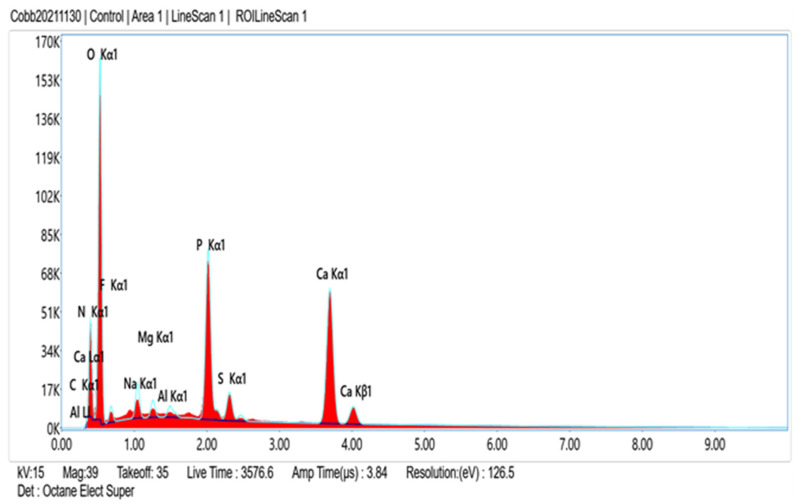
Conversion of the line scan seen in Figure 9 (SRP only specimen) to an EDS spectrum shows relative peak height of the various elements of interest. Dominate peaks are O, P, and Ca and lesser peaks for N, F, Na, Mg, and S. Peaks for Al and C are likely derived from the carbon sputter coating and aluminum mounting stub.

**Figure 11 dentistry-11-00022-f011:**
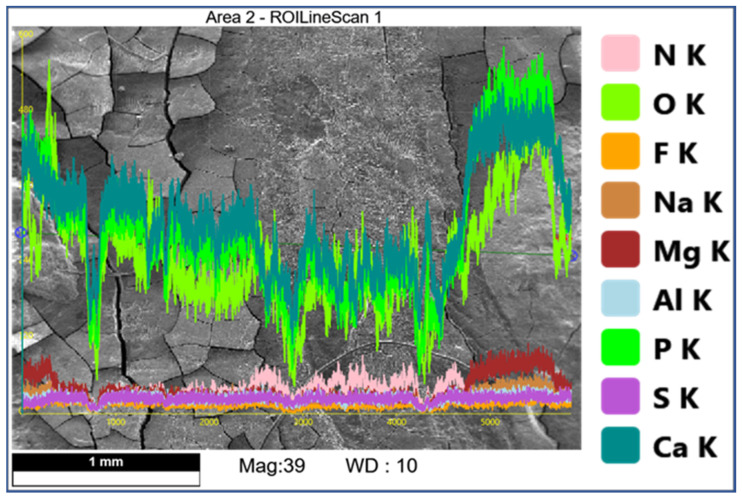
SEM of SRP + EDTA specimen with overlay of color-coded EDS line scan showing peaks (from baseline) for F, S, N, Na, Mg, Ca, P, and O. Compared to Figure 9, this scan shows obvious differences in location and presence of N, Na, and Mg. The scan started and stopped on calculus deposits with an intervening area of SRP in between. The scan traversed a line of ≈3 mm in length. Original mag. 39×; Bar = 1 mm. Working distance of 10 mm.

**Figure 12 dentistry-11-00022-f012:**
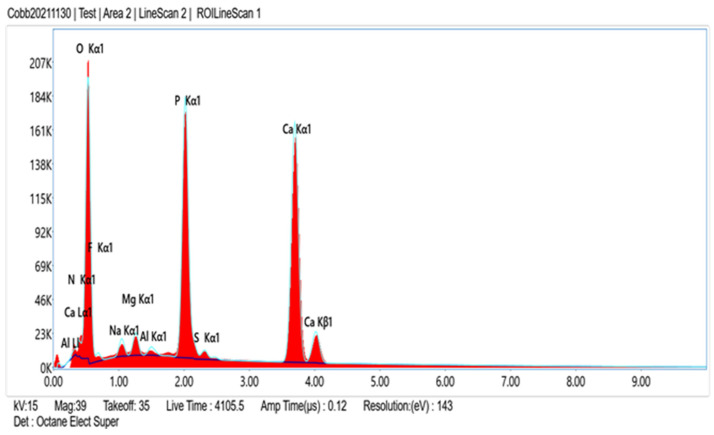
Conversion of the line scan seen in Figure 11 to an EDS spectrum shows relative peak height of the various elements of interest. As with the SRP-only specimen (Figure 10), the dominate peaks are O, P, and Ca, with lesser peaks for N, F, Na, Mg, and S.

**Figure 13 dentistry-11-00022-f013:**
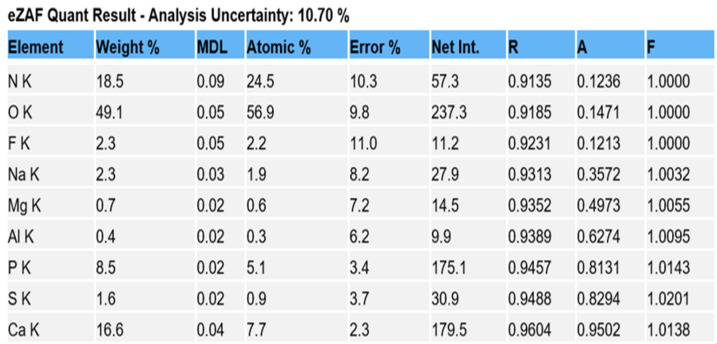
Data from SRP-only specimen generated for the elements of interest, expressed as Weight % and Atomic %. Both represent the relative concentration of the element in the sample. Atomic % is calculated by dividing the element Weight % by its atomic weight. Note the analysis uncertainty of 10.70% indicating the error of measurement which is influenced by the irregular topography of the root surface.

**Figure 14 dentistry-11-00022-f014:**
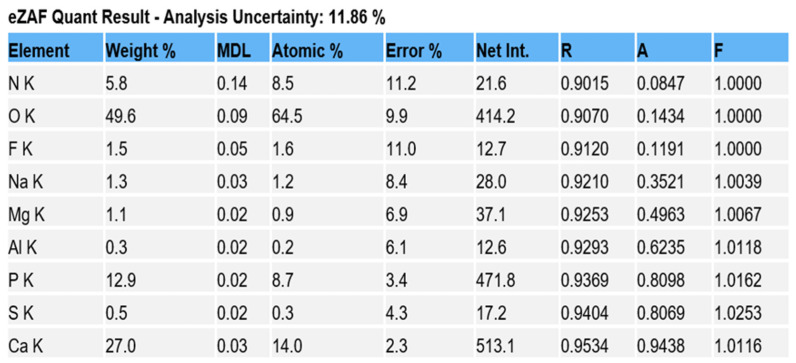
Data generated for the elements of interest, expressed as Weight % and Atomic %. Note the analysis uncertainty of 11.86%. Compared to data in Figure 13, there are decreases in Weight % and Atomic % or N, F, Na, and S and increases in Mg, P, and Ca. Oxygen was nearly equal in both specimens (49.1 for SRP only vs. 49.6 for SRP + EDTA).

## Data Availability

Supporting data for this study can be obtained upon request from the corresponding author.

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
