# Peer review of "Effect of EDTA Gel on Residual Subgingival Calculus and Biofilm: An In Vitro Pilot Study"

_dentistry, 2023, doi:10.3390/dj11010022_

Round 1
Reviewer 1 Report
Tables and charts are self explanatory
Wonderful study.
Author Response
No response necessary. On behalf of the authors, I extend my thanks to the reviewer for taking the time to read the manuscript. Having been a peer reviewer for numerous manuscripts I full appreciate the time and effort.
Thank you.
Reviewer 2 Report
Major comments
Overall this is a poor study with no clinical significance. There are just a few specimens, it is descriptive, no statistics and as such a poor design. There is no reference or speculation or study on the use of EDTA in clinical dental practice. Is that safe, is it for daily use, can the dentist apply it? And then, is there any need for it? Traditional scaling and root planing works well and periodontitis can be arrested with this simple mechanical intervention.
I think the study was performed more than 10 years ago. The authors must ensure they used fresh, and not out of date EDTA (beyond expiration date) EDTA prefgel. I say this because the authors acquired Prefgel from Biora, which company does not exist anymore for at least 10 years as it was bought by another company and there after again sold to yet another company.
Moreover, specimens were stored in formalin, for how long? 1 week, 1 month, 1 year, 10 years? What is the effect of formalin storage on residual calculus? We need appropriate controls without formalin pretreatment.
What was the whole purpose of EDS? The clinical relevance is not explained. The data analysis is on 2 specimens, why is it important? The discussion is filled for 75% with a repetition of these results.
In the Conclusions I expect the answer on the “So what” question: specify the importance and clinical relevance of the work.
Specified comments
Title: add “An in vitro study”
Introduction:
- Start with an intro on which disease we are talking about; at least the Abstract mentions periodontitis, but not the introduction.
- Remove bullits at the end of Intro, why you have it?
- You write in second bullit: “to significantly alter the physical properties”. Please explain how, what is altered and why would that be of significance, please answer the question: so what.
Mat & Meth
- Explain when the study was performed: dates and years. I think the study was performed more than 10 years ago.
- Explain explicitly that you have used fresh, and not out of date EDTA (beyond expiration date) prefgel. I say this because the authors acquired Prefgel from Biora, which company does not exist anymore in the last 10 years at least as it was bought by another company and there after again sold to yet another company.
- Explain: The treated root surface was 4 x 4 mm: but later and in results the authors analyse an area of 48 sq mm. How is this possible??
- Specimens were stored in formalin, explain and specify for how long? 1 week, 1 month, 1 year, 10 years? What is the effect of formalin storage on residual calculus? We need appropriate controls without formalin pretreatment.
- Spelling Philips
- What is the whole purpose of EDS? Explain, what is the clinical relevance? The data analysis is on 2 specimens, why is it important? The discussion is filled for 75% with a repetition of these results, why? Why were certain elements of interest?
Results
- In the descriptive text often it is unclear to which specimens you refer, SRP alone or with EDTA treatment. Please improve/specify
- In each figure add arrows and perhaps with different colors, to really draw the attention to a specified area. Figure 1: I do not see calculus.
- Improve each figure legend, e.g. figure 2, 3, which specimens?
- Figure 5: where can I see smearing or smudging, explain it to the reader
- And so on about the figures
- Figure 9 spelling traversed. Figures 9 and 11: why is there a “K” behind element. What is the meaning of it.
- Transform figures 13 and 14 to self-made tables and explain everything, all abbreviations and all parameters and explain what the reader should tale from it.
Discussion
- Why too much emphasis on EDS results. These are unclear and have no clinical meaning, at least it has never been explained.
- I expect a critical review on the clinical effectiveness of SRP, and with that, why there might be a need for application of EDTA gel.
Conclusions
- I expect in the Conclusions the answer on the “So what” question: specific the importance and relevance of the work.
Author Response
- Overall this is a poor study with no clinical significance.
With all due respect, the reviewer is entitled to his opinion but two of the authors are American Board certified periodontists with almost 100 years, collectively, of experience in clinical periodontics, teaching and research. We happen to think this study does have clinical significance.
2. There are just a few specimens, it is descriptive, no statistics and as such a poor design.
The authors are aware of the limited number of specimens. To ensure the reader is also made aware of this fact, we have inserted a paragraph following Line 299 in the discussion that reads as follows: Although the current investigation is a pilot study presenting descriptive results, limitations must be considered for design of future efforts. The major limitation is the small sample size which does not allow for statistical analysis of the EDS data. An additional consideration would be expansion of the experimental design to include other biologic surface modifiers for comparison to EDTA.
In addition, the manuscript title as been changed to read: The Effect of EDTA Gel on Residual
Subgingival Calculus and Biofilm: An In Vitro Pilot Study
A change in title accomplishes two things: 1) In vitro is more descriptive of the project, and 2) the addition of pilot study explains the limited number of specimens.
The comment of “no statistics and as such a poor design” is not worthy of comment except to note that the literature is replete with descriptive studies that have no statistical data.
- There is no reference or speculation or study on the use of EDTA in clinical dental practice. Is that safe, is it for daily use, can the dentist apply it? And then, is there any need for it?
The following paragraph was added following Line 242. This paragraph addresses the issue of EDTA use in dental practice (at least in periodontics). It should be noted that 24% EDTA was originally used to acid etch dentin/enamel before insertion of composite restorative materials. This paragraph also addresses the issue of safety. Is there a need for EDTA, that is why the study was done.
“Partially due to the antimicrobial nature of EDTA and the etching effect on mineralized surfaces, a 24% EDTA gel has been used as a surface modifier in periodontal root coverage surgeries, e.g., coronally advanced flaps, subepithelial connective tissue grafts, modified coronally advanced tunnel procedure, etc., with only modest benefit. It should be noted, however, regardless of any benefit derived from the use of EDTA, clinical trials have reported no adverse effects. [17,18]”
- Traditional scaling and root planing works well and periodontitis can be arrested with this simple mechanical intervention.
I would ask the reviewer to read the following article. We can then discuss his/her dogmatic claim that SRP “works well”. Cobb CM, Sottosanti JS. A re-evaluation of scaling and root planing. Journal of Periodontology. 2021;92:1370–1378. DOI: 10.1002/JPER.20-0839. It is this paper that initiated the authors curiosity about the impact of EDTA on micro-islands of residual calculus that are not visible to the unaided human eye. Further, there is considerable evidence that any residual calculus, regardless of volume, is capable of sustaining an inflammatory response.
- I think the study was performed more than 10 years ago. The authors must ensure they used fresh, and not out of date EDTA (beyond expiration date) EDTA prefgel. I say this because the authors acquired Prefgel from Biora, which company does not exist anymore for at least 10 years as it was bought by another company and there after again sold to yet another company.
The study was done during the months of July 2022 to October 2022. The reviewer is correct regarding the PrefGel and Biroa. That was our mistake. The PrefGel was actually purchased new from Straumann and this mistake has been corrected (see Line 94).
6. Moreover, specimens were stored in formalin, for how long? 1 week, 1 month, 1 year, 10 years? What is the effect of formalin storage on residual calculus? We need appropriate controls without formalin pretreatment.
There is nothing in the histologic/SEM/EDS literature suggesting that 10% neutral buffered formalin loses its fixation effect with age, unless of extreme age (> 5 years). As the reviewer can see, the SEMs are of high quality and high magnification, figures 3 and 6 being 10,000x. Obviously, the formalin had no adverse impact on the microbes that coat the calculus surface. Secondly, one of the authors is a recognized expert in surface analysis of biologic specimens and assures that formalin, even specimens stored in formalin for a short period (< 1 year) has little impact on EDS scan results. If, however, one was to use Raman Spectroscopy or FTIR, formalin may impact specimens at the molecular level.
Lastly, non-fixed or unstabilized specimens are not suitable for high resolution SEM and EDS scans. Therefore, the reviewers request that nonfixed specimens be used as controls cannot be fulfilled.
- What was the whole purpose of EDS? The clinical relevance is not explained. The data analysis is on 2 specimens, why is it important? The discussion is filled for 75% with a repetition of these results.
The reviewer needs to read the paper again. There are multiple published studies that used EDS to study non-diseased and diseased tooth root surfaces. Clinical relevance is discussed in the introduction (Lines 33-60). The discussion section is long and somewhat meticulous to take into consideration readers who may not be familiar with EDS.
- In the Conclusions I expect the answer on the “So what” question: specify the importance and clinical relevance of the work.
The following sentence has been added to the Conclusions: EDTA mediated disruption of biofilm microbes and their EPM and the smear layer resulting from root surface instrumentation may enhance SRP and decrease periodontal inflammation.
- Title: add “An in vitro study”
As noted in #2 (above), In Vitro has been added to title.
- Start with an intro on which disease we are talking about; at least the Abstract mentions periodontitis, but not the introduction.
Line 34 of the introduction has been changed to read: “…. videoscopes, when used to treat periodontitis, resulted in …...”
- Remove bullets at the end of Intro, why you have it?
Bullets have been removed. They were not in the original manuscript when submitted.
- You write in second bullet: “to significantly alter the physical properties”. Please explain how, what is altered and why would that be of significance, please answer the question: so what.
I would respectfully request the reviewer read the cited reference (#3). The introduction is not the appropriate section to present a detail discussion of the literature. That remains for the discussion.
- Explain when the study was performed: dates and years. I think the study was performed more than 10 years ago.
See #5 above for explanation.
- Explain explicitly that you have used fresh, and not out of date EDTA (beyond expiration date) prefgel. I say this because the authors acquired Prefgel from Biora, which company does not exist anymore in the last 10 years at least as it was bought by another company and there after again sold to yet another company.
See #5 above for explanation.
15. Explain: The treated root surface was 4 x 4 mm: but later and in results the authors analyse an area of 48 sq mm. How is this possible?
The sentence at Lines ?? was changed to read: “For the 3 SEM specimens, an aggregate total root surface area of 48 mm2 was assessed with 12 sites per treated root surface being examined, i.e., 36 sites.”
Hopefully, this will make clear what was done. There is no conflict between the two statements. One refers to the treated surface area (4 x 4 mm) and the other refers to the aggregate surface area examined by the SEM (48 mm2).
- Specimens were stored in formalin, explain and specify for how long? 1 week, 1 month, 1 year, 10 years? What is the effect of formalin storage on residual calculus? We need appropriate controls without formalin pretreatment.
This is a repeat question. See #6 above.
17. Spelling Philips
Line 100: changed spelling to Philips.
18. What is the whole purpose of EDS? Explain, what is the clinical relevance? The data analysis is on 2 specimens, why is it important? The discussion is filled for 75% with a repetition of these results, why? Why were certain elements of interest?
With all due respect, the authors feel they have explained in detail the reason for the EDS analysis and are not going to address these questions again.
- In the descriptive text often it is unclear to which specimens you refer, SRP alone or with EDTA treatment. Please improve/specify
Line 147 changed to read: SRP only. With the exception of Line 147, all other statements use the term SRP only or SRP + EDTA.
- In each figure add arrows and perhaps with different colors, to really draw the attention to a specified area. Figure 1: I do not see calculus.
Inserted two yellow arrows pointing to calculus an d changed legend to read: …. calculus (yellow arrows) and several cavitation-like defects in root surface (white arrows).
- Improve each figure legend, e.g. figure 2, 3, which specimens?
Figure 2 now reads: Untreated root surface showing calculus …….
Figure 3 now reads: Untreated root surface at high magnification showing biofilm ……..
- Figure 5: where can I see smearing or smudging, explain it to the reader
Figure 5 now contains a white rectangle outline the smear layer.
- Figure 9 spelling traversed. Figures 9 and 11: why is there a “K” behind element. What is the meaning of it.
Spelling was corrected to read “traversed”. The following was inserted in the legend: “The K following each element designation refers to the electron shell closest to the nucleus.”
- Transform figures 13 and 14 to self-made tables and explain everything, all abbreviations and all parameters and explain what the reader should tale from it.
The authors consider this request as totally unnecessary to two reasons. First, it is simply an exercise to reinvent the wheel. Second, converting to a self-made table increases risk of error in data entry. By using the table as shown on the SEM/EDS screen ensures no chance of falsifying data.
- Why too much emphasis on EDS results. These are unclear and have no clinical meaning, at least it has never been explained.
The authors consider this a statement of opinion and, therefore, will not address the issue.
- I expect a critical review on the clinical effectiveness of SRP, and with that, why there might be a need for application of EDTA gel.
If the reviewer truly wants a review on the clinical effectiveness of SRP, I would refer him/her to the following papers: Cobb CM, Sottosanti JS. A re-evaluation of scaling and root planing. Journal of Periodontology. 2021;92:1370–1378; Cobb CM. Non-Surgical periodontal therapy: Mechanical. Annals of Periodontology. 1996;1:443-490.
As the reviewer must know, it is not the purpose of this manuscript to present a review of SRP. One must assume the reader has some baseline knowledge regarding non-surgical periodontal therapy or why else would they choose to read the article?
- I expect in the Conclusions the answer on the “So what” question: specific the importance and relevance of the work.
This has been previously addressed in #8 above.
Reviewer 3 Report
The aim of the present investigation was to assess effect of EDTA gel on subgingival calculus and biofilm in vitro.
General Comment: The article is in line with the journal subject, but some parts of manuscript should be improved. The investigation is interesting and the present paper is recommended for publication in the present journal after minor revision.
Title - please consider adding information about type of paper (in vitro study).
Line 36-38 - "Microislands of calculus and are not visible with 3.5X surgical loupes but can be identified by fluorescence when using a 655-nm diode laser [3] and visualized at 40X using a videoscope [3,4]" - after calculus there should be something or "and" should be erased ?
Line 219 - In comparing wight % values for the elements of interest in the SRP only (Fig. 13) - here is a typo.
The limits of the present study should be described in discussion part (e.g. small number of samples).
Please add a short paragraph in discussion part about usage of EDTA in periodontology (e.g. root conditioning before CAF or SCTG, MCAT - like here: PMID: 30293186 and PMID: 34431001).
Author Response
- Title - please consider adding information about type of paper (in vitro study).
Title has been changed to be more descriptive and to address the limited number of specimens. The title now reads: The Effect of EDTA Gel on Residual Subgingival Calculus and Biofilm: An In Vitro Pilot Study
- Line 36-38 - "Microislands of calculus and are not visible with 3.5X surgical loupes but can be identified by fluorescence when using a 655-nm diode laser [3] and visualized at 40X using a videoscope [3,4]" - after calculus there should be something or "and" should be erased ?
Line 37: Deleted the word “and”.
- Line 219 - In comparing wight % values for the elements of interest in the SRP only (Fig. 13) - here is a typo.
Line 219: Changed spelling to “weight”.
- The limits of the present study should be described in discussion part (e.g., small number of samples).
Following Line 299 the following new paragraph has been Inserted: Although the current investigation is a pilot study presenting descriptive results, limitations must be considered for design of future efforts. The major limitation is the small sample size which does not allow for statistical analysis of the EDS data. An additional consideration would be expansion of the experimental design to include other biologic surface modifiers for comparison to EDTA.
- Please add a short paragraph in discussion part about usage of EDTA in periodontology (e.g., root conditioning before CAF or SCTG, MCAT - like here: PMID: 30293186 and PMID: 34431001).
Following Line 242 the following new paragraph has been Inserted: Partially due to the antimicrobial nature of EDTA and the etching effect on mineralized surfaces, a 24% EDTA gel has been used as a surface modifier in periodontal root coverage surgeries, e.g., coronally advanced flaps, subepithelial connective tissue grafts, modified coronally advanced tunnel procedure, etc., with only modest benefit. It should be noted, however, regardless of any benefit derived from the use of EDTA, clinical trials have reported no adverse effects. [17,18]
Please note the addition of new references #17 & #18 required renumbering of reference thereafter. Also, original references #21 and #25 were removed so the total number of references remains at 30.
Reviewer 4 Report
This is an important paper. The significance of residual calculus to continued periodontal inflammation has been long overlooked. Your use of EDS clearly pointed out the differences between SRP alone and SRP followed by EDTA.
Please state which author(s) performed the therapy and their experience levels. PrefGel is distributed by Straumann not Biora (line 94). Line 219 weight not wight. Line 228 infect not infection of.
Author Response
- Please state which author(s) performed the therapy and their experience levels.
Line 92: Following the last sentence of the paragraph, the following was inserted: All SRP was performed by a periodontist (SKH) with 45+ years of clinical and teaching experience.
- PrefGel is distributed by Straumann not Biora (line 94).
Line 94: Deleted (PrefGel™, Biora AB, Malmo, Sweden) and replaced with (PrefGel™, Straumann, Basel, Switzerland)
- Line 219 weight not wight.
Line 219: Changed spelling to weight.
- Line 228 infect not infection of.
Line 228: Should read: … contribute to the infection of the periodontal pocket.
Round 2
Reviewer 2 Report
I have no further comments.
Reviewer 3 Report
The article has been corrected taking into account all my comments.
Reviewer 4 Report
Nice job